# Small-Scale Urban Object Anomaly Detection Using Convolutional Neural Networks with Probability Estimation

**DOI:** 10.3390/s23167185

**Published:** 2023-08-15

**Authors:** Iván García-Aguilar, Rafael Marcos Luque-Baena, Enrique Domínguez, Ezequiel López-Rubio

**Affiliations:** 1Department of Computer Languages and Computer Science, University of Málaga, Bulevar Louis Pasteur 35, 29071 Málaga, Spain; rmluque@uma.es (R.M.L.-B.); enriqued@lcc.uma.es (E.D.); elr@uma.es (E.L.-R.); 2Biomedical Research Institute of Málaga (IBIMA), C/Doctor Miguel Díaz Recio 28, 29010 Málaga, Spain

**Keywords:** anomaly detection, convolutional neural network, super-resolution

## Abstract

Anomaly detection in sequences is a complex problem in security and surveillance. With the exponential growth of surveillance cameras in urban roads, automating them to analyze the data and automatically identify anomalous events efficiently is essential. This paper presents a methodology to detect anomalous events in urban sequences using pre-trained convolutional neural networks (CNN) and super-resolution (SR) models. The proposal is composed of two parts. In the offline stage, the pre-trained CNN model evaluated a large dataset of urban sequences to detect and establish the common locations of the elements of interest. Analyzing the offline sequences, a density matrix is calculated to learn the spatial patterns and identify the most frequent locations of these elements. Based on probabilities previously calculated from the offline analysis, the pre-trained CNN, now in an online stage, assesses the probability of anomalies appearing in the real-time sequence using the density matrix. Experimental results demonstrate the effectiveness of the presented approach in detecting several anomalies, such as unusual pedestrian routes. This research contributes to urban surveillance by providing a practical and reliable method to improve public safety in urban environments. The proposed methodology can assist city management authorities in proactively detecting anomalies, thus enabling timely reaction and improving urban safety.

## 1. Introduction

The recent decrease in the price of video surveillance systems has led to an exponential increase in their installation on several roads and urban areas. Processing the information collected by these systems is a critical task because it includes detecting anomalous events to ensure security and improve reaction times by the relevant authorities. Anomaly detection involves identifying events that do not constitute expected activities or behaviors. This is a complex problem, since the captured sequences present several issues. First of all, obtaining a dataset in which labeled anomalies are present requires great effort, and sometimes, it may be possible that there are no anomalies in a particular area. On the other hand, it is important to highlight the complexity of the anomaly to be detected. For example, a pedestrian crossing a pedestrian area does not present any anomaly. However, that is not the case if the pedestrian is walking on a road, for example. In addition, video surveillance systems are placed in high places, which sometimes makes it difficult to detect where the anomalous event exists. Detecting this type of event efficiently and autonomously is critical for tasks such as security, recognition, and monitoring in urban roads or autonomous systems applied to driving.

Anomaly prediction approaches were initially based on several classical techniques and strategies used for anomaly identification. Among them are feature-based methods, such as calculating the trajectory of the element to be analyzed to detect possible changes in its behavior through its direction. Other techniques include clustering to identify pedestrians in common areas, extracting similar behavior patterns, and identifying those that deviate from this standard as a threat. Despite the existence of these techniques, there are still shortcomings in this area that need to be improved. Several strategies have emerged based on deep learning using object detection models. However, there are still shortcomings when it comes to identifying anomalies based on the nature of the image, since in cases where the element is small and difficult to locate, the anomaly will not be detected. Another significant problem is the diversity related to the spatial location of the elements to be analyzed and their appearance. Some of the object identification models that stand out in this field are, for example, FASTER R-CNN [1], among others.

This paper presents a methodology applied to autonomously detecting anomalies in sequences, using convolutional neural networks (CNN) and image quality enhancement techniques, such as super-resolution, to improve safety on urban roads. The proposed method aims to autonomously learn the behavior of the elements appearing in the sequence to recognize and categorize the elements identified as anomalies. In this case, anomaly detection is focused on vehicles and pedestrians captured by a video surveillance system. The main contributions are the following:The creation and collection of a synthetic dataset to test the proposed methodology. This set consists of urban sequences where pedestrians and vehicles of variable dimensions transit. The training set is composed of sequences to learn the typical patterns of these elements, while sequences conform to the test set with multiple anomalies to be identified.Applying the described methodology using several pre-trained object detection and super-resolution models to improve the elements’ mean average precision (mAP).Testing and evaluation of the described technique using various metrics.

The rest of the article is organized as follows. Section 2 presents related works. Section 3 presents the proposed methodology. Subsequently, Section 4 presents the selected dataset and the metrics and results. Section 5 outlines the discussions. Finally, Section 6 presents the conclusions and future directions.

## 2. Related Work

For the proposed methodology, object detection models and super-resolution models are considered to identify anomalies effectively. Several related works in these areas are cited in this discussion. Finally, relevant works related to anomaly identification are also presented in this discussion.

### 2.1. Convolutional Neural Network Models

Detecting elements is essential for identifying possible anomalies based on the proposed methodology. Therefore, according to the domain where anomaly identification using object detection is desired, selecting which type of convolutional neural network should be applied is essential. Deep learning advancements have considerably enhanced element detection compared to classical techniques. Currently, two main categories of models are based on their approach to performing element identification.

The first approach involves identifying regions of interest through a region proposer and identifying elements within these areas. This model type offers the advantage of achieving a higher mean average precision (mAP) when identifying elements. However, the time required to perform such inference increases considerably. Within this group, standout several models obtain good results according to the application scope. These models are precisely evaluated in the proposed methodology to identify anomalies. The first approach, denoted as CenterNet [2], presents an efficient solution, as it first explores the visual patterns within each region that composes the input image. Other models, such as Faster R-CNN [1], introduce the concept of a Region Proposal Network (RPN), a fully convolutional network that predicts object scores and boundaries, resulting in reduced execution time during inference. Finally, it is important to highlight models such as EfficientNet [3]. This family of models highlights mainly due to the focus on scalability and computational efficiency since it uses scaling methods that balance the model’s depth, width, and resolution. This results in improved performance without significantly affecting computational complexity. A base network architecture is developed to search for the optimal model composition, and it is then scaled to create a family of models ranging from the simplest (B0) to the most complex (B7).

In the second group, some models prioritize speed in inference times sacrificing accuracy. Notably, one of the highlighted models in this category is SSD (Single-Shot MultiBox Detector) [4]. SSD employs a unique convolutional neural network architecture that directly performs detection and classification in a single pass, unlike region-based approaches. It stands out as a model evaluated alongside the ones mentioned earlier.

In the proposed methodology for anomaly detection, pre-trained models from both categories were used to enhance element detection for identifying elements in the synthetic dataset of urban sequences.

### 2.2. Super-Resolution Models

Increasing the image input size is one of the most common techniques to improve the performance given by the object detection models. Upscaling the input size of the image also increases the number of pixels that compose each object. By increasing the dimensions of the input image, the number of pixels representing each object is also amplified, thereby facilitating their identification using object detection models. This upscaling process is accomplished through super-resolution (SR) algorithms, which rely on convolutional neural networks to generate enlarged images as output. The use of SR techniques aids in preserving image quality while effectively improving object detection capabilities, making it crucial to identify and choose the most appropriate model from the ones listed in this context.

The combination of super-resolution and convolutional neural networks has shown to be an effective approach for enhancing object detection accuracy. Iván García Aguilar et al. [5], in their research, aimed at improving vehicle detection in road image sequences, employing a 2× scale super-resolution technique around previously detected objects. The objective is to enable new detections of small objects in a subsequent pass of the detection model. The premise of its development is based on using the detections defined by the model initially to be used as tentative areas on which to re-infer. As a result, this approach yields favorable outcomes, significantly improving accuracy without re-training or modifying the original detection model.

The Super-Resolution Generative Adversarial Network (SRGAN) has been a significant advancement in generating realistic textures for single image super-resolution. However, the generated details sometimes come with undesirable artifacts. Models such as SRCNN can be highlighted in which a network is proposed to improve detection in multispectral remote images. Jiang et al. [6] noted in their work that traditional super-resolution models fail to generate enlarged images with edge detail in noise-contaminated images, thereby complicating the detection of objects in satellite images. In an attempt to solve this problem, they propose a new SR architecture based on GAN networks, called EEGAN, which is invariant to noise. Another example of using super-resolution based on GAN networks is stated in [7], where a new multi-scale augmentation method known as Scale Adaptive Image Cropping (SAIC) is proposed, which consists of cropping the input image in different ways to apply super-resolution to varying scales according to the estimated object size to improve detections in low-resolution aerial imagery obtained from UAVs.

To further enhance visual quality, Wang et al. propose a new model denoted as Enhanced SRGAN (ESRGAN) as a result of improving network architecture, adversarial, and perceptual loss. This model achieves better visual quality with more realistic and natural textures compared to other models. These authors extend the powerful ESRGAN to a practical restoration application, obtaining a model called Real-ESRGAN [8]. This model is trained using synthetic data to simulate complex real-world degradations. It also addresses common ringing and overshoot artifacts in these images. A U-Net discriminator with spectral normalization is employed to enhance discriminator capability and stabilize training dynamics, resulting in superior visual performance compared to other models.

The methodology employs pre-trained super-resolution models, such as Real-ESRGAN [8], to enhance object detection accuracy. This pre-trained model, which has been fine-tuned on synthetic data to simulate real-world degradations, significantly improved the visual quality of the input images, making it easier for the pre-trained object detection models to identify and locate objects accurately in the urban sequences.

### 2.3. Anomaly Detection

Numerous works have addressed anomaly detection using classical techniques, primarily relying on reconstructive or discriminative approaches [9,10,11,12]. These methods aim to learn the normal behavior patterns within a specific application domain. However, they often encounter limitations in effectively capturing complex distributions in video sequences.

Due to the increasing demand for security and safety, anomaly detection has gained significant interest in intelligent video surveillance analysis. Ref. [13] proposed a novel method for anomaly detection in pedestrian conduct. The approach utilizes motion-appearance features and dynamic behavior changes over time, employing Locality Sensitive Hashing (LSH) functions for detection. Key contributions include robust pedestrian segmentation, the Dynamics of Pedestrian Behavior (DoPB) feature, and the Adaptive Anomaly Weight (AAW) with block-based optical flow tracking, demonstrating effectiveness in detecting and localizing anomalies. Irina et al. propose an automated deep learning-based anomaly detection technique called DLADT-PW (Deep Learning based Anomaly Detection Technique in Pedestrian Walkways) [14] to enhance pedestrian safety. The traditional manual examination of abnormal events in video surveillance systems is cumbersome, making an automated surveillance system essential for computer vision researchers. DLADT-PW uses preprocessing to remove noise and enhance image quality. The detection process involves the use of the Mask Region Convolutional Neural Network (Mask-RCNN) [15] with Densely Connected Networks (DenseNet). The DLADT-PW model aims to detect and classify anomalies in pedestrian walkways, such as cars, skating, and jeeps.

The use of Convolutional Neural Networks applied to anomaly identification has led to works such as the one proposed by Xing Hu et al. [16], in which a weakly supervised framework is proposed for the detection and localization of abnormal behavior in scenes, using object detection with Faster R-CNN, behavior description with a Large Scale Optical Flow Histogram (HLSOF) descriptor and classification with a Multiple Instance Support Vector Machine (MISVM). Ref. [17] presents a novel approach for unsupervised pedestrian anomaly event detection by leveraging trajectory localization and prediction. Unlike conventional reconstruction-based methods, the proposed framework utilizes prediction errors of normal and abnormal pedestrian trajectories to detect spatial and temporal anomalies. The experimental results on real-world benchmark datasets demonstrate the effectiveness and efficiency of the trajectory-predictor-based anomaly detection pipeline in identifying anomalous activities of pedestrians in videos across varying timescales. In work such as [18], a proposed deep learning model for abnormal behavior detection uses YOLOv3 object detection technology to detect pedestrians, followed by a hybrid Deep-SORT algorithm to track pedestrians and obtain tracking trajectories. In addition, a convolutional neural network (CNN) is used to extract the action features of each tracked trajectory, and a short-term memory network (LSTM) is used to build an anomalous behavior identification and prediction model. Ref. [19] presents an automatic anomaly detection model based on hierarchical social hunting optimization and a deep convolutional neural network (HiS-Deep CNN) for surveillance videos, including object detection and tracking. However, one of the problems with some of these methodologies is that sufficient information must be acquired for training these models. The methodology presented in this article, therefore, makes use of synthetic information generated through the CARLA simulator [20]. B. Sophia et al. [21] presents a novel Panoptic Feature Pyramid Network-based Anomaly Detection and Tracking (PFPN-ADT) model for pedestrian walkways in video surveillance. The model’s primary objective is to recognize and classify anomalies in pedestrian walkways, such as vehicles and skaters. The proposed approach utilizes the Panoptic Feature Pyramid Network (PFPN) for object recognition and the Compact Bat Algorithm (CBA) with Stacked Auto Encoder (SAE) for object classification, demonstrating the enhanced performance of the PFPN-ADT technique in detecting anomalies effectively.

### 2.4. Differences with Other Proposals

The proposed approach focuses on anomaly detection at the agent behavior level, specifically for detecting vehicles and pedestrians with anomalous behavior in traffic videos. Anomaly detection at the raw pixel level is unsuitable for the application, as it would not differentiate between individual objects and would only detect deviations from the background image. A convolutional neural network detects the agents and obtains their high-level object information, such as class labels and bounding boxes. Applying multidimensional data analysis techniques to the image data would not yield the desired results, as it would only identify deviating pixels without providing object-level information. Similarly, an autoencoder would detect individual pixels but fail to capture the necessary high-level object behavior information.

Several systems designed for pixel-level anomaly detection in still images were unsuitable for agent behavior analysis in traffic videos. While these systems could identify moving pixels, they could not detect whole objects and their corresponding high-level properties. Thus, to effectively detect anomalous agent behavior, the proposed approach processes the incoming video using a convolutional neural network to detect agents and then carries out anomaly detection in the space of possible high-level object behaviors, providing a focused and accurate analysis of vehicles and pedestrians violating traffic rules.

The proposed methodology differs from the related works in its approach to anomaly detection. While the other methods focus on utilizing motion-appearance features, trajectory localization or specific deep learning models for pedestrian anomaly detection, the methodology combines super-resolution techniques with convolutional neural networks for object detection. This novel approach aims to enhance object detections by leveraging a training sequence and creating a matrix of common object locations (bounding boxes). During the evaluation phase, the percentage of deviations from these common regions is calculated to identify anomalies, specifically in cars and pedestrians. By incorporating super-resolution and convolutional neural networks, the methodology offers a unique perspective on improving object detection and detecting anomalies in video surveillance scenarios.

## 3. Models and Methods

In this section, the proposed methodology denoted as SR-DAI (Super-Resolution and Detection with Anomaly Identification) is described in detail. The provided method is shown in Figure 1. The methodology is designed to improve object detection in video sequences by leveraging a two-step process. Initially, a training dataset is used. Super-resolution enhances the quality of each frame to create an image tilling. These areas with the upscaling factor are then used for re-inference, improving detection results from the object detection model. Once the translation of the box to the real coordinates system is performed, the clustering operation identifies simultaneous detections for the same object to delete it. Subsequently, a density matrix is computed based on the detected bounding boxes to identify the common locations of the elements. During the validation phase, each frame undergoes super-resolution and is fed to the object detection model. By comparing the bounding box positions with the previously calculated density matrix, the methodology determines whether an element can be classified as an anomaly, enabling more accurate and effective anomaly detection in video surveillance. Below, each of the components comprising the proposed methodology is detailed.

### 3.1. Convolutional Neural Networks for Object Detection

The presented methodology includes using a pre-trained object detection model, providing a solid basis for determining the improvement in anomaly detection. This section will present a general description of how object detection models using Convolutional Neural Networks (CNNs) work, followed by a detailed explanation of each evaluated model, highlighting their features and capabilities.

Convolutional Neural Networks (CNNs) are a specialized class of deep learning models widely used for computer vision tasks, particularly object detection. These models are designed to automatically learn and extract meaningful features from images, enabling them to identify and localize objects accurately. The basis of CNNs is the convolutional layers, which perform a mathematical operation called convolution, which consists of sliding small filters (also known as kernels) over the input image to detect local patterns, such as edges, textures, and shapes. The convolution operation is represented as:(1)$$F\left(i,j\right)=\sum_{m}\sum_{n}I\left(i+m,j+n\right)\times K\left(m,n\right)$$
where F(i,j) is the element in the resulting feature map, *I* is the input image and *K* is the convolutional filter. The double summation $\sum_{m}\sum_{n}$ indicates that all elements of the filter *K* and the selected local region of the input image *I* at position (i,j) are summed. The convolution operation results in a feature map highlighting relevant patterns and features in the input image.

After the convolution operation, a non-linear activation function is applied element-wise to the feature map. Multiple types of these non-linear operations exist, such as RELU, which is applied element-wise to the feature map, allowing the model to capture complex relationships between image features. Next, pooling layers are used to reduce the spatial dimensions of the feature maps and retain essential information. Max-pooling is a common pooling technique that selects the maximum value within a local region of the feature map. This downsampling process reduces the computational complexity and makes the model more robust to object position and scale variations.

The resulting feature maps are flattened into a one-dimensional vector following several convolutional and pooling layers. This vector is then fed into fully connected layers acting as classifiers. The output of the fully connected layer is obtained by:(2)$$y=W_{\mathrm{fc}}x+b_{\mathrm{fc}}$$
where *x* represents the flattened feature vector, $W_{\mathrm{fc}}$ denotes the weights and $b_{\mathrm{fc}}$ represents the biases of the fully connected layer. The class scores are then processed through an activation function such as softmax to obtain the probabilities for each object class. Bounding box regression is also employed to accurately predict the bounding box coordinates that cover the detected objects. The general operation of convolutional neural networks (CNNs) applied to object detection has already been described. Five models were evaluated to test the effectiveness of the proposed methodology. The performance and unique characteristics of each model considered for evaluation are presented below:CenterNet HourGlass104 Keypoints 1024 × 1024: CenterNet Keypoints is a one-stage object detection framework that efficiently predicts object centers and regresses the bounding box size. The HourGlass104 backbone effectively captures multi-scale features. In addition to detecting object bounding boxes, this variant predicts keypoints associated with each object, making it suitable for tasks such as human pose estimation. The HourGlass architecture uses repeated down-sampling and up-sampling stages, allowing it to capture fine-grained details while maintaining a global context.CenterNet HourGlass104 1024 × 1024: This CenterNet variant omits the keypoint prediction branch, focusing only on object detection. Using the HourGlass104 backbone, it retains the advantages of multi-scale feature representation and precise object localization. The absence of the keypoint prediction branch reduces the model’s computational complexity, making it more efficient.Faster R-CNN Inception ResNet V2 1024 × 1024 (RetinaNet152): Faster R-CNN is a two-stage object detection model that separates region proposal generation and object classification. The Inception ResNet V2 backbone provides good feature extraction capabilities. The first stage generates region proposals using a Region Proposal Network (RPN), which efficiently proposes candidate object bounding boxes. In the second stage, these proposals are further refined and classified to produce the final detections.EfficientDet D4: EfficientDet is a scalable and efficient object detection model that balances accuracy and computational efficiency. The EfficientDet D4 variant is optimized to detect objects at different scales with high accuracy. It leverages a composite scaling method that uniformly scales the model’s depth, width and resolution.SSD ResNet152 V1 FPN 1024 × 1024 (RetinaNet152): SSD (Single-Shot Multibox Detector) is a one-stage object detection model that directly predicts object categories and bounding boxes at multiple scales. The ResNet152 V1 FPN backbone incorporates feature pyramid networks (FPN) for multi-scale feature extraction and improved performance. The FPN helps the model handle objects of several sizes effectively.

The evaluated models are all pre-trained on the COCO (Common Objects in Context) dataset [22]. This dataset is widely used for training and evaluating object detection models. These models are available in Tensorflow 2 Model Zoo repository (https://github.com/tensorflow/models/blob/master/research/object_detection/g3doc/tf2_detection_zoo.md, accessed on 1 July 2023).

### 3.2. Convolutional Neural Networks for Super-Resolution

The Convolutional Neural Network (CNN) model used for super-resolution in this study is Real-ESRGAN. This model is an advanced deep learning architecture explicitly designed for single-image super-resolution to enhance image resolution and visual quality.

Real-ESRGAN employs a generator network that takes a low-resolution image as input and outputs a high-resolution version of the same image. The generator network is based on the Enhanced Super-Resolution Generative Adversarial Network (ESRGAN) architecture, which incorporates residual blocks to facilitate learning and capture intricate image details effectively. The generator network in Real-ESRGAN can be represented as follows:(3)$$G\left(I_{\mathrm{LR}}\right)=I_{\mathrm{SR}}$$
where $I_{\mathrm{LR}}$ denotes the low-resolution input image and *G* represents the generator network. The output $I_{\mathrm{SR}}$ is the high-resolution version of the input image, which is expected to exhibit enhanced visual quality and finer details. To achieve good results, Real-ESRGAN leverages a pre-trained discriminator network. The discriminator network, denoted as D, assesses the realism of the generated high-resolution images. The adversarial loss between the generated images and the real high-resolution images is computed to train the generator network effectively. The discriminator is a U-Net discriminator with spectral normalization to increase discriminator capability and stabilize the training dynamics.

### 3.3. Methodology

Once the object detection and super-resolution models have been specified, the methodology presented is detailed as follows.

Given a sequence S, the first step is to extract each of the frames that compose it:(4)$$S=\left\{\left(I_{l}\right)\;|\;l\in \left\{1,\ldots ,N\right\}\right\}$$

Consider an input image, denoted as $I_{l}$, an object detection model denoted as D, and a super-resolution model, set as Z, which allows improving the full resolution of the D image with an upscaling factor of ×2. Initially, the image $I_{l}$ is processed by the object detection model D ([Object detection] in Figure 1), thus obtaining the first list of initial detections:(5)$$D\left(I\right)=L_{ini}$$
(6)$$L_{ini}=\left\{\left(\alpha_{i},\beta_{i},\gamma_{i},\delta_{i},\lambda_{i},\rho_{i}\right)\;|\;i\in \left\{1,\ldots ,N\right\}\right\}$$
where *N* stands for the number of detections, the coordinates of the top left corner of the *i*-th detection within *Y* are noted $\left(\alpha_{i},\beta_{i}\right)\in R^{2}$, the coordinates of the bottom right corner of the *i*-th detection within *Y* are noted $\left(\gamma_{i},\delta_{i}\right)\in R^{2}$, $\lambda_{i}$ stands for the detected class label and $\rho_{i}\in R$ denotes the obtained class score. As $\rho_{i}$ increases, the confidence in the existence of an object of class $\lambda_{i}$ at that detection also increases.

This initial detection is performed to detect the largest objects in the image, which are more easily identifiable by the model D and more likely to be sliced into different subimages. Then, the image $I_{l}$ is divided into several subimages $S_{I}^{i}$ ([Image Tiling] in Figure 1). This transformation is done with the aim that, after applying super-resolution to each subimage, the super-resolved image $Z(S_{I}^{i})$ will match as best as possible the maximum allowed input size, noted as size(D), of the object detection model D. Please note that the model D adjusts the image to its predefined input size by applying bicubic interpolation. This interpolation results in worse image quality with a negative effect on object detection performance. The previously described subimage size selection is given by:(7)$$\mathrm{size}\left(S_{I}^{i}\right)=\arg\min_{\mathrm{size}\left(W\right)}∥z\cdot \mathrm{size}\left(W\right)-\mathrm{size}\left(D\right)∥$$
where *z* denotes the upscaling factor to be applied. At the same time, in this step, the relative position of each subimage $S_{I}^{i}$ according to the original image I is saved for subsequently undoing the transformation of the detections of each subimage. To avoid detecting the same object several times across different subimages, each of them is extended by the inner sides to obtain an overlapping of the bounding boxes between objects distributed among two or more subimages, thus allowing to unify the detections in a later step. It is worth mentioning that this technique can be extrapolated to any other magnification factor *z*.

For each of the generated subimages, super-resolution will be applied. The goal of this step ([TTA] in Figure 1) is to present to the detection model the same image but in different shapes to increase the number of detections in the image. Therefore, each of these new augmented images is passed through the object detector D, which in turn generates a list of detections, called $L_{SR}$: (8)$$L_{SR}=\left[a_{1},a_{2},\ldots ,a_{n}\right]$$

These contain the annotations $a_{i}$ that store the information related to the detection performed by the D model. Let *a* be a tuple of the form:(9)a=(b,c,s)
where *s* represents the score, given by the model to the performed detection, *c* the class estimated by the model and whose value is predetermined by the COCO standard and *b* is a list that contains four elements that correspond to the coordinates of the four vertices of the rectangular bounding box that locates the object in the image [Object Detection on TTA] in Figure 1:(10)$$b=[y_{min},x_{min},y_{max},x_{max}]$$

Based on *b* and the relative position of each tile or subimage $S_{mathcalI}^{i}$ within the original image I, by undoing the applied transformation, it becomes possible to globally locate each detected object in the image [Translate Coordinates] in Figure 1, which in the case of super-resolution would be as follows:(11)$$b^{\prime }=\left(x_{i},y_{i}\right)+\frac{1}{z}b$$
where $\left(x_{i},y_{i}\right)$ are the coordinates of the upper left corner of the subimage $\left(x_{i},y_{i}\right)$ of the original image and *z* is the applied upscaling factor, which in the case of the presented proposal is 2. Having grouped them in the same coordinate system of the image I, they are unified in a final list of annotations where, ideally, each annotation $a_{i}$ matches only one object detected in the input image I. For this purpose, a clustering method ([Clustering] in Figure 1) is employed. This consists of grouping the input list L into a list of clusters $L_{K}=\left[k_{1},k_{2},\ldots ,k_{n}\right]$, where each cluster contains the annotations that are similar to each other in the sense that their feature vectors (b,c,s) are close enough to be considered the repeated detection of the same object.
(12)$$k=\left\{a_{i},a_{j}\in L/IoU\left(b_{i},b_{j}\right)\geq threshold_{IoU}\wedge c_{i}=c_{j}\right\}$$

Therefore, the clustering criterion between each pair of annotations $a_{i},a_{j}$ is based on the closeness between their respective bounding boxes. If the IoU of both is greater than a threshold value, called $threshold_{IoU}$, the objects are grouped into the same cluster. Finally, a list of final annotations is returned, where the annotation of each cluster with the highest score given by the detection model $L_{out}$ is obtained.

The proposal works at the agent behavior level, not the raw data (pixel) sample level, so it does not intend to identify anomalous pixel values concerning the rest of the pixel values. This would not work for the application at hand, namely the detection of agents (vehicles and pedestrians) with anomalous behavior. In other words, the goal is to detect which vehicles and pedestrians cross areas of the scene that are forbidden for them, as opposed to vehicles and pedestrians that move according to traffic rules. If anomaly detection were applied at the data sample (pixel) level, it would detect deviations from the background image, i.e., the background of the traffic scene. This would detect the raw foreground pixels belonging to all vehicles and all pedestrians without detecting the individual objects and irrespective of the anomaly of the high-level behavior of the agents to which those pixels belong. Therefore, anomaly detection at the data sample (pixel) level is inappropriate for this application. Anomaly detection must be done on the distribution of agent behaviors and not on the raw data sample (pixel value) distribution to detect agents of anomalous behavior. Therefore, the first step is to process the incoming video with an object detection subsystem (a convolutional neural network) that detects the agents and yields their behavior (the agent classes and their bounding boxes). Then, anomaly detection is carried out on the space of those possible high-level object behaviors. It must be noted that this space is not high dimensional, so dimensionality reduction is not necessary.

Once the annotations have been obtained after applying the methodology, the next step is estimating the probability density of elements identified at each point of the image [Density Estimation Calculation] in Figure 1. For this purpose, a kernel-based probability density estimator is used. A kernel function smooths the data and estimates the probability density in a continuous space. Considering a set of points with coordinates (x,y), which represent the location of the detected elements in the frame, the probability density at each point is estimated as a function of its neighborhood.

Subsequently, the kernel-based probability density estimator is used to generate a matrix representing the zones through which the elements of each class pass. The formula of the kernel-based density estimator used to estimate the probability density $\hat{f}\left(x,y\right)$ at each location (x,y) of the image is as follows:(13)$$\hat{f}\left(x,y\right)=\frac{1}{nh^{2}}\sum_{i=1}^{n}K\left(\frac{\left(x,y\right)-\left(x_{i},y_{i}\right)}{h}\right)$$

The formula considers the total number *n* of sample elements $\left(x_{i},y_{i}\right)$ in the class, the bandwidth *h* that controls the smoothing of the estimate, and the kernel function *K* that defines the shape and contribution of each neighboring element. By applying this formula for each point in the image, a probability density matrix is obtained that reflects the class distribution in the image. Subsequently, a threshold is set on the probability density to identify areas where the presence of elements of the class is unusual, allowing anomalies to be detected. The choice of the parameters of the density estimator, such as bandwidth and kernel function, will depend on the characteristics of the data and the problem at hand.

Once the matrix with the most common zones for the various classes to be analyzed in a training sequence has been obtained, the methodology mentioned above is applied with the super-resolution approach to online sequences. Once the elements that compose it have been identified, the distribution in the matrix previously calculated is checked to determine whether or not it corresponds to an anomaly.

## 4. Experiments and Results

Below, the selected dataset is presented, along with the evaluation metrics established to verify the robustness of the methodology proposed. Finally, the results obtained are shown.

### 4.1. Dataset

Anomaly detection is a significant challenge in the development of multiple fields, such as autonomous driving or safety in video surveillance systems, because it involves the identification of unusual situations that may represent a safety risk—in this case, for pedestrians or vehicles moving through a busy area. In real-world scenarios, anomalies are unusual compared to typical situations. This can make it difficult to collect sufficient anomaly data in a real dataset, thus limiting the ability to train and evaluate anomaly detection models effectively.

Therefore, synthetic dataset such as the CARLA (Comprehensive Autonomous Driving Simulator) simulator play a crucial role. It is a powerful simulator designed for the development and training applied to autonomous driving systems. This simulator facilitates the creation of synthetic scenarios, thus allowing the exhaustive control of anomalous events that may arise in a particular sequence. This enables more extensive and balanced dataset to be obtained in terms of anomaly detection.

It offers a variety of scenarios that allow one to simulate an environment realistically. Within these scenarios, it is possible to modify weather, the time of day in which the sequence has been taken, lighting conditions and pedestrian and vehicle traffic, among others. Thus, it provides a versatile environment for evaluating the presented methodology. Examples of scenarios are shown in Figure 2 and Figure 3.

In addition, it provides tools for the automatic generation of annotated sequences. The simulator automatically labels objects and events of interest based on requirements previously established by the user based on the generated sequence. Within these annotations, information related to the location of the identified elements is included, as well as the class to which the elements belong, such as vehicles, pedestrians, etc. Thus, it facilitates the creation of datasets with complex requirements, thus eliminating the problem of tedious and erroneous manual labeling and allowing the generation of sequences appropriate for the given application environment. Figure 4 shows an automatic annotated frame. Finally, the advantage offered by this simulator is the ability to generate anomalous events, such as collisions or pedestrian crossing in no-trespassing areas. These events are crucial for evaluating the robustness of the presented methodology under several conditions.

### 4.2. Metrics

An approach has been established that combines the generation of sequences with many pedestrians and vehicles to determine the most transit areas for these classes and evaluate the anomaly detection performance of the presented methodology. Subsequently, the described methodology is applied to sequences where a series of anomalies exist, thus allowing to evaluate the effectiveness of the presented proposal. To quantify it, two types of quantitative metrics have been established.

First, the average accuracy of the detections obtained by the model is evaluated. This metric is commonly used for the evaluation of object detection algorithms. It is mainly based on the accuracy of the detections identified by the network. For this purpose, precision–recall curves are generated for each class to be evaluated, subsequently calculating each class’s area under the curve (AP). By averaging the various APs for the classes to be evaluated, the mean accuracy (mAP) is obtained. A high mAP indicates that the model can identify the elements appearing in the sequence accurately and effectively. Therefore, it is of vital importance to improve this measure to detect possible threats that cannot be identified by the model initially.

Secondly, the number of anomalies detected is evaluated. For this purpose, a manual count of anomalies in a particular frame is performed. This metric provides us with an additional measure that allows us to evaluate the performance of the methodology presented and its ability to identify and locate the various anomalies that may be present in the sequence. The greater the number of anomalies detected, the greater the capacity of the system to find anomalous events and respond effectively to them.

### 4.3. Results

The results obtained from evaluating the robustness of the proposed approach using established quantitative measures are detailed below. These measures include the mean Average Precision (mAP) and the number of anomalies detected. Table 1 presents the parameters that were used to conduct the tests. Specifically, the minimum confidence required to consider an element as a positive detection is specified, along with the anomaly threshold used to determine whether an element is anomalous based on density. Additionally, the scaling factor employed for the super-resolution of the images is provided. These values were thoughtfully selected to enhance the performance of the proposed approach and ensure consistent and reliable results.

The IoU (Intersection over Union) threshold value was set to 0.25, allowing for the inclusion of possible detections by the neural network on objects with very small sizes. Regarding the anomaly threshold, it was set to 0.5. A lower value might increase false positives, while a higher value could result in overlooking some challenging-to-detect anomalies in the scene. The scaling factor was chosen as 2, striking an appropriate balance between improving image resolution and avoiding considerable size distortion of elements in the region. A much higher magnification factor could lead to incorrect class inference by the neural network.

In Table 2, a comparison is made between the mean accuracy (mAP) obtained by the original model (RAW) and the presented proposal (OURS). The comparison of different object detection models reveals the superiority of the proposed approach in detecting pedestrians and cars in various scenarios. Notably, using the EfficientDet D4 model stands out as it consistently outperforms other models in terms of mean Average Precision (mAP) across different evaluation metrics and sequences.

In the case of the pedestrian class, a substantial increase in the mAP is observed. While the RAW model achieves an accuracy of 1.8% for the first sequence, the presented proposal manages to increase this percentage to 12.8% without retraining or modifying the object detection model. The results are not limited to this class, as a consistent improvement is observed in the other evaluated category, corresponding to the car class. In this class, the improvement is even more pronounced. In Sequence 2, the CenterNet HourGlass104 model obtained an mAP of 14.2%, outperforming the base model’s mAP of 0.86% by 13.34%.

Thanks to the application of this technique, all evaluated models have achieved notable improvements in element identification, leading to a significant enhancement in mAP scores. By incorporating the super-resolution approach using a pre-trained model, the resolution of surveillance images captured from elevated points has been effectively increased, surpassing the limitations associated with detecting small-sized objects. This enhanced resolution, combined with pre-trained object detection models, has resulted in more accurate and reliable detection of pedestrians and vehicles in complex surveillance scenarios.

To correctly determine the density in the location of the elements in the scene according to their class, improving the ability to detect them is vital. Applying the RAW model, many elements went unnoticed, thus generating an underestimation of the densities. Given the improved accuracy, the calculation is much more reliable, thus obtaining a better understanding of the spatial distribution of the elements according to the class to which they belong.

The heat maps based on the calculated densities are presented in Figure 5 and Figure 6. For each figure, the heat map calculated with the RAW model is shown on the left side, and the one obtained after applying the presented proposal is on the right side. Figure 5, which corresponds to calculating the probability densities for the elements belonging to the pedestrian class, shows how the original model fails to correctly detect the elements located in the background. However, after applying the proposal, more accurate detection is achieved, correctly identifying the location of these elements in the heat map. In Figure 6, the heat maps of the second scene, focused on the car class, are shown. Again, it can be seen how the calculation of probability densities is much more accurate and detailed when applying the presented proposal since traffic areas that were not initially detected are recognized. This improvement results in a more complete and reliable analysis of the probability density of these elements in the scene.

Table 3 presents the results related to the number of anomalies present in the evaluated sequences and those detected by applying the RAW model versus the presented proposal. The table highlights the number of anomalies contained in each scene and the number of anomalies detected using both the RAW model and the proposed approach (OURS) for each sequence and model. The results demonstrate that the proposed methodology consistently outperforms the RAW model in detecting anomalies across all evaluated object detection models. In Sequence 1, the proposed approach (OURS) significantly improves the anomaly detection capabilities for all models, as indicated by the higher number of anomalies detected compared to the RAW model.

For example, with the CenterNet HourGlass104 Keypoints 1024 × 1024 model in Sequence 1, the RAW model detected 243 anomalies. At the same time, the proposed approach (OURS) managed to identify a more substantial number of anomalies, reaching 335. Similarly, with the Faster R-CNN Inception ResNet V2 1024 × 1024 model, the RAW model detected only 57 anomalies, whereas the proposed approach detected 222. This remarkable improvement in anomaly detection across different models showcases the effectiveness of the proposed methodology in identifying potential anomalies with greater accuracy.

In Sequence 2, the trend continues, with the proposed approach (OURS) consistently surpassing the RAW model in anomaly detection for all evaluated models. For instance, with the EfficientDet D4 model, the proposed approach (OURS) identified 81 anomalies, whereas the RAW model does not detect anything, thus demonstrating the effectiveness and potential of the same in this situation where they could initially go unnoticed.

As a qualitative result, Figure 7 and Figure 8 are presented, where the anomalies detected by both the RAW model and the presented proposal are highlighted in red color, while those not considered as such are shown in green color. The difference in anomaly detection between the two approaches can be visualized by analyzing the figures. As previously demonstrated by quantitative data, the proposal can identify a greater number of elements. The elements identified as anomalies are fewer in number and, in some cases, could go undetected. This difference in detection highlights the effectiveness of the presented approach to provide a more complete and accurate view of the anomalies in the scene.

## 5. Discussion

The proposed methodology for anomaly detection in surveillance scenarios has been thoroughly evaluated, and this section discusses the key findings. The investigation focused on the effectiveness of various object detection models in conjunction with a pre-trained super-resolution model to enhance anomaly identification. The results presented in the earlier sections demonstrate significant improvements in anomaly detection compared to the base models.

Using pre-trained object detection models, including CenterNet, Faster R-CNN, EfficientDet D4 and SSD, has proven to be a critical factor in achieving better results. These models exhibited superior generalization and resistance to input distribution shifts by leveraging the pre-trained weights obtained from the COCO dataset.

Combined with the pre-trained object detection models, incorporating a super-resolution technique further enhanced the proposed methodology’s performance. The number of pixels constituting objects increased by applying super-resolution to the input images before feeding them into the object detection model. This process significantly facilitated the identification of small or partially occluded objects, resulting in improved accuracy and recall. Consequently, the proposed methodology demonstrated a remarkable ability to detect anomalies that the base object detection models would have missed.

Quantitative evaluation of the proposed methodology using mean average precision (mAP) and the number of anomalies detected revealed compelling results. For example, in Sequence 2, the RAW model using EfficientDet D4 does not detect anything, while the presented proposal identified a remarkable 81 anomalies. This represents an astounding improvement in anomaly detection, showcasing the methodology’s robustness and effectiveness. Moreover, comparing mAP values between RAW and OURS for different object classes and sequences consistently indicated the superior performance of the proposed methodology.

However, despite these promising results, the methodology has limitations that warrant discussion. One limitation lies in the reliance on pre-trained models. While these models offer strong generalization capabilities, domain-specific factors may influence their effectiveness. In scenarios with adverse lighting conditions or unique object classes not adequately represented in the pre-training dataset, the performance of the proposed methodology may be impacted. Further research is needed to assess the robustness of the methodology in diverse real-world surveillance environments.

The proposed methodology’s computational complexity should also be considered, especially when deploying the system in real-time surveillance applications. Integrating super-resolution with object detection may demand substantial computational resources, potentially limiting its real-time applicability on resource-constrained devices. Therefore, efforts should be made to optimize the methodology without compromising its accuracy and effectiveness.

Furthermore, the proposed methodology was evaluated on synthetic video sequences generated using the CARLA simulator. Although these sequences mimic real-world scenarios to some extent, the model’s performance in actual surveillance footage needs to be validated. Real-world data often introduce more complexities, such as variations in lighting, weather conditions and object appearances, which can impact the methodology’s performance.

## 6. Conclusions and Future Work

The proposed methodology, which involves inferring on super-resolved areas of the image to identify more elements and calculate more accurately the density estimation, has demonstrated its efficacy in increasing the number of pixels that constitute objects, leading to improved identification by the object detection model. The results obtained from applying this methodology on two synthetic video sequences generated using the CARLA simulator have shown a significant advancement in anomaly detection within the scenes. Applying the proposed approach enables the accurate identification of anomalies that the base model would have missed or misclassified.

The results obtained from evaluating the proposed methodology against the base object detection models have consistently shown notable improvements in element identification and a significant enhancement of mAP scores. For example, the proposed approach (OURS) using the EfficientDet D4 model achieved a remarkable improvement in mAP scores for the pedestrian class in Sequence 1, with a value of 12.8% compared to the base model’s score of 1.8%. These compelling results underscore the practical utility and robustness of the proposed methodology in enhancing surveillance and security systems. The system achieves more reliable and precise anomaly detection by effectively combining super-resolution and object detection techniques. When using the CenterNet HourGlass104 Keypoints 1024 × 1024 model on Sequence 2, the proposed approach (OURS) successfully detected 84 anomalies, while the RAW model only identified 4 anomalies.

However, further research and development are warranted to explore the applicability of the proposed methodology in scenarios with adverse lighting conditions. Incorporating techniques such as LUT (Look Up Table) to enhance image illumination could benefit complex environments with challenging lighting situations. This would improve the methodology’s performance in diverse real-world scenarios and extend its usability in various surveillance applications. Moreover, the proposed methodology’s versatility warrants investigation into its effectiveness when applied to different domains and types of objects. Exploring its adaptability to diverse object classes and real-world surveillance environments would provide valuable insights into the system’s generalizability and broaden its potential applications.

## Figures and Tables

**Figure 1 sensors-23-07185-f001:**
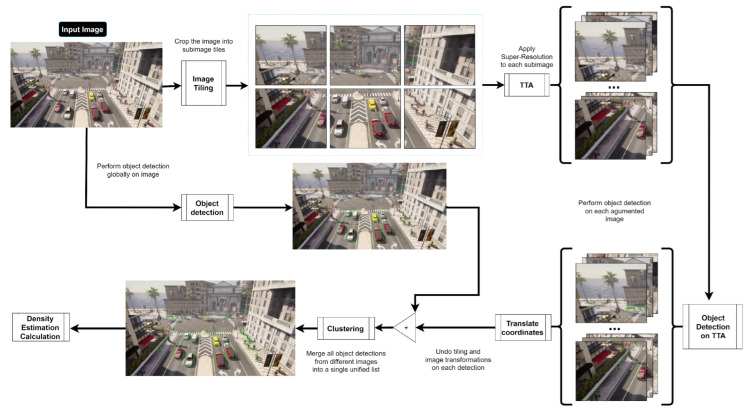
Workflow of the proposed technique denoted as SR-DAI. The bounding boxes in green represent the elements identified by the object detection model.

**Figure 2 sensors-23-07185-f002:**
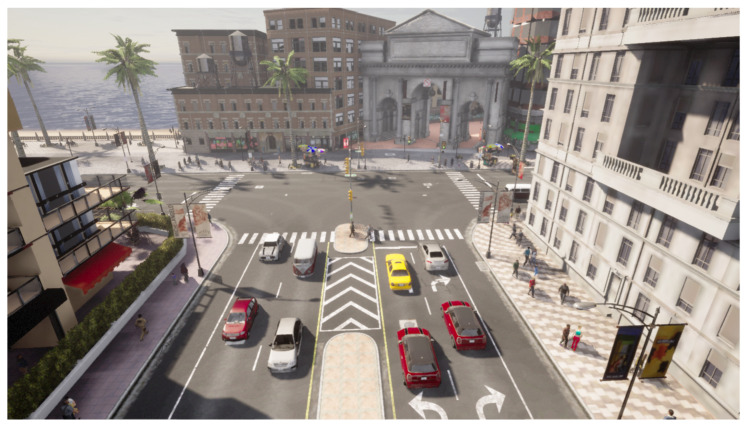
An example of the first scenario created.

**Figure 3 sensors-23-07185-f003:**
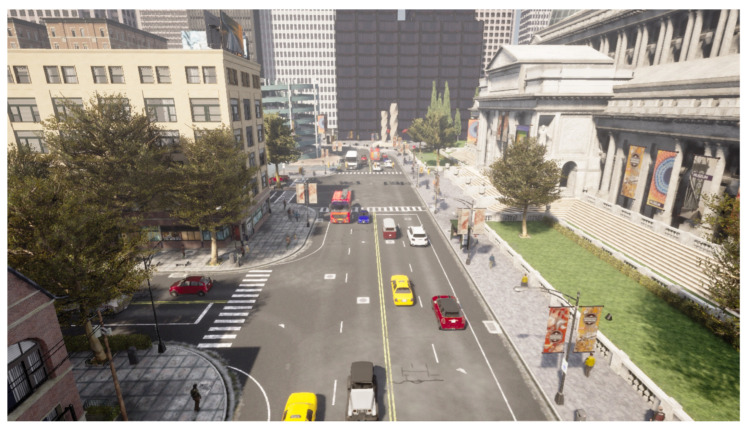
An example of the second scenario created.

**Figure 4 sensors-23-07185-f004:**
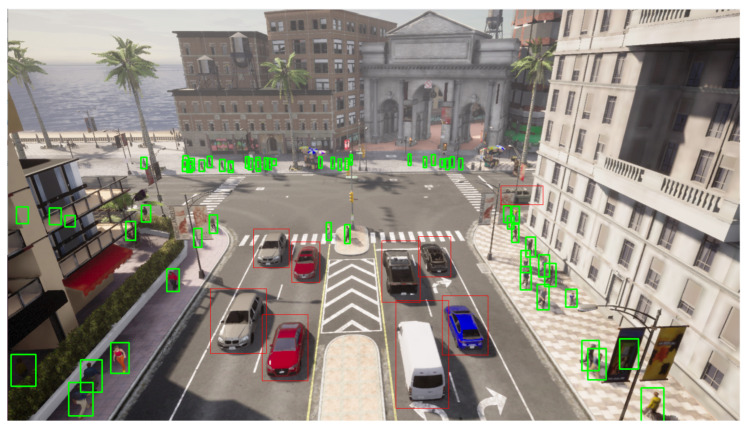
An example of an automatically annotated frame.

**Figure 5 sensors-23-07185-f005:**
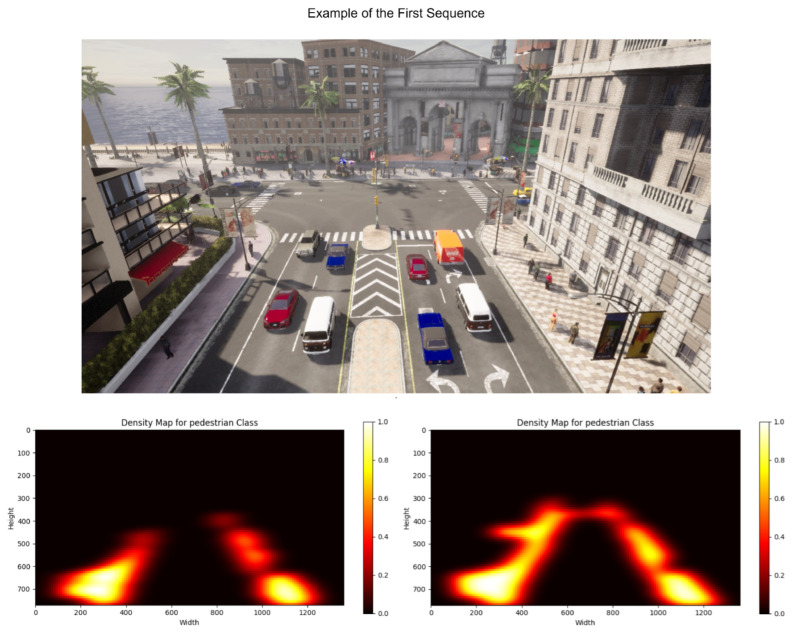
Example of the heat maps calculated for the first sequence. On the left is the one obtained using the RAW model, while on the right is the one calculated based on the presented proposal.

**Figure 6 sensors-23-07185-f006:**
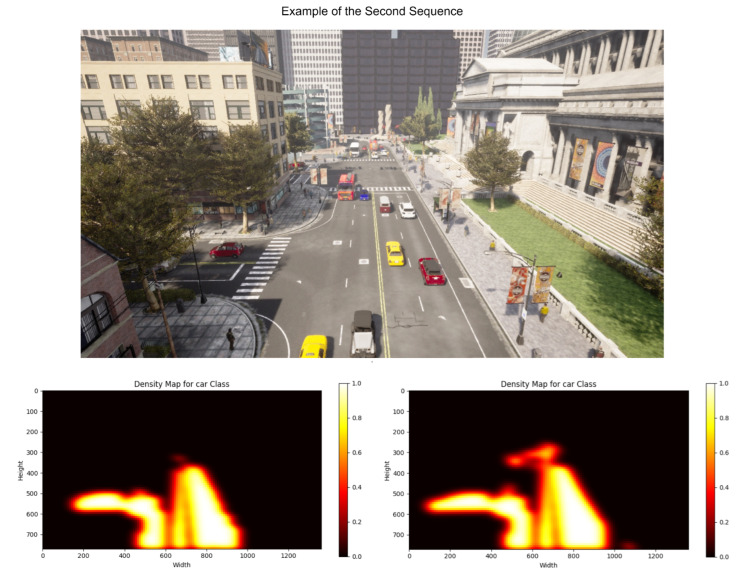
Example of the heat maps calculated for the second sequence. On the left is the one obtained using the RAW model, while on the right is the one calculated based on the presented proposal.

**Figure 7 sensors-23-07185-f007:**
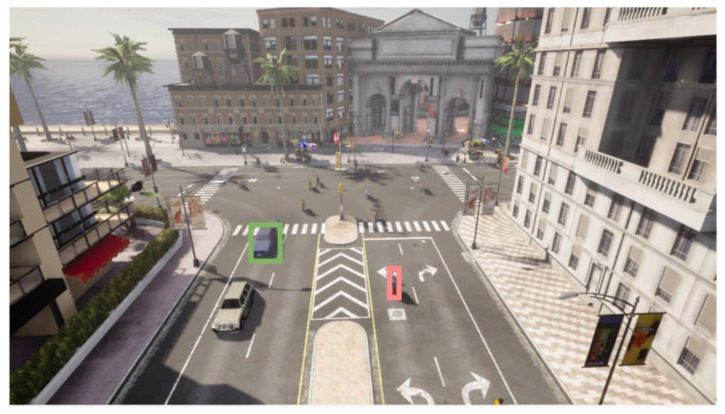
Visualization of the anomalies identified by the RAW model at the top and the proposal presented at the bottom for one frame to the first sequence. Anomalies are represented in red, while non-anomalous objects are represented in green.

**Figure 8 sensors-23-07185-f008:**
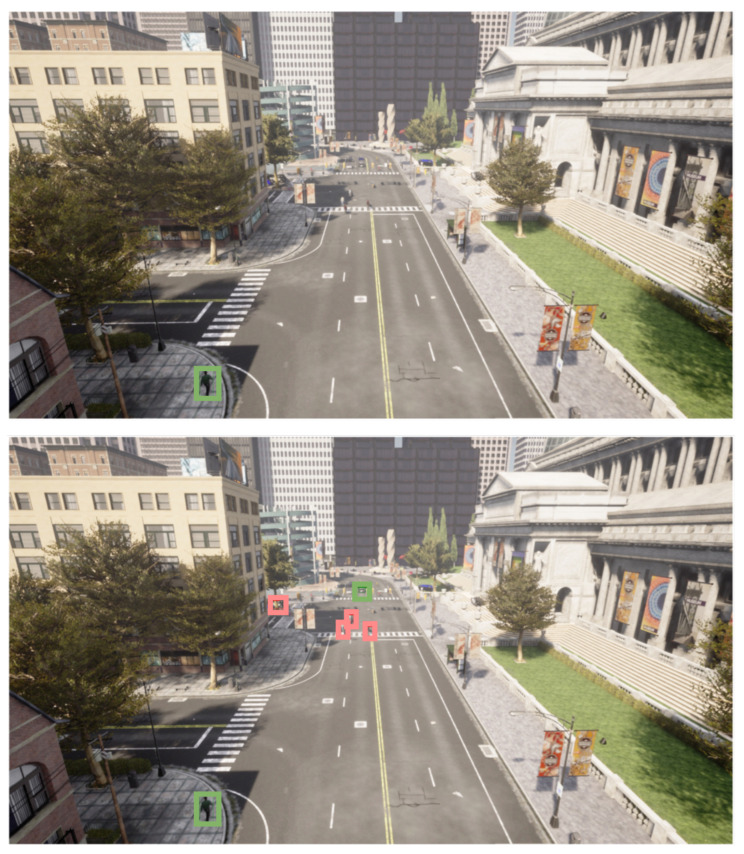
Visualization of the anomalies identified by the RAW model at the top and the proposal presented at the bottom for one frame to the second sequence. Anomalies are represented in red, while non-anomalous objects are represented in green.

**Table 1 sensors-23-07185-t001:** Selected values of the hyper-parameters.

Hyper-Parameter	Value
Selected Model	Efficientdet D4
IoU Threshold	0.25
Anomaly Threshold	0.5
Tiling Factor	2

**Table 2 sensors-23-07185-t002:** Results obtained for the two synthetic sequences generated. RAW denoted the mAP obtained by the base model and on OURS is the mAP obtained by the presented proposal (higher is better). The best results are marked in **bold**. The evaluated classes are pedestrians and cars.

		Pedestrian Class				Car Class			
		**Sequence 1**		**Sequence 2**		**Sequence 1**		**Sequence 2**	
**Model**	**Metric**	**RAW**	**OURS**	**RAW**	**OURS**	**RAW**	**OURS**	**RAW**	**OURS**
CenterNet HourGlass104 Keypoints 1024 × 1024	IoU = 0.50:0.95|area = all	0.087	**0.119**	0.038	**0.056**	0.197	**0.237**	0.109	**0.158**
	IoU > 0.50|area = all	0.272	**0.346**	0.095	**0.150**	0.386	**0.449**	0.278	**0.374**
	IoU > 0.75|area = all	0.028	**0.045**	0.018	**0.024**	0.120	**0.180**	0.030	**0.051**
	IoU = 0.50:0.95|area = Small	0.081	**0.101**	0.033	**0.051**	0.190	**0.494**	0.020	**0.058**
	IoU > 0.50|area = Medium	0.114	**0.203**	0.208	**0.237**	0.184	**0.236**	0.225	**0.292**
CenterNet HourGlass104 1024 × 1024	IoU = 0.50:0.95|area = all	0.081	**0.122**	0.043	**0.056**	0.186	**0.241**	0.086	**0.142**
	IoU > 0.50|area = all	0.259	**0.351**	0.106	**0.149**	0.389	**0.457**	0.256	**0.365**
	IoU > 0.75|area = all	0.024	**0.044**	0.025	**0.026**	0.093	**0.171**	0.024	**0.034**
	IoU = 0.50:0.95|area = Small	0.077	**0.105**	0.036	**0.052**	0.345	**0.497**	0.010	**0.054**
	IoU > 0.50|area = Medium	0.105	**0.203**	0.208	**0.231**	0.180	**0.235**	0.194	**0.268**
Faster R-CNN Inception ResNet V2 1024 × 1024	IoU = 0.50:0.95|area = all	0.033	**0.071**	0.097	**0.130**	0.015	**0.033**	0.055	**0.068**
	IoU > 0.50|area = all	0.112	**0.228**	0.255	**0.299**	0.033	**0.087**	0.212	**0.219**
	IoU > 0.75|area = all	0.008	**0.017**	0.045	**0.097**	0.010	**0.017**	0.016	**0.018**
	IoU = 0.50:0.95|area = Small	0.024	**0.055**	0.150	**0.157**	0.008	**0.027**	0.010	**0.020**
	IoU > 0.50|area = Medium	0.086	**0.157**	0.102	**0.140**	0.151	**0.207**	0.126	**0.147**
EfficientDet D4	IoU = 0.50:0.95|area = all	0.018	**0.128**	0.010	**0.052**	0.133	**0.226**	0.062	**0.124**
	IoU > 0.50|area = all	0.051	**0.350**	0.018	**0.114**	0.262	**0.417**	0.147	**0.296**
	IoU > 0.75|area = all	0.007	**0.047**	0.010	**0.034**	0.092	**0.211**	0.014	**0.040**
	IoU = 0.50:0.95|area = Small	0.010	**0.104**	0.006	**0.042**	0.258	**0.346**	0.013	**0.040**
	IoU > 0.50|area = Medium	0.073	**0.249**	0.151	**0.268**	0.115	**0.206**	0.138	**0.255**
SSD ResNet152 V1 FPN 1024 × 1024 (RetinaNet152)	IoU = 0.50:0.95|area = all	0.007	**0.085**	0.137	**0.211**	0.010	**0.035**	0.022	**0.100**
	IoU > 0.50|area = all	0.020	**0.260**	0.301	**0.446**	0.017	**0.083**	0.067	**0.253**
	IoU > 0.75|area = all	0.003	**0.021**	0.083	**0.143**	0.010	**0.021**	0.007	**0.032**
	IoU = 0.50:0.95|area = Small	0.008	**0.069**	0.007	**0.235**	0.006	**0.029**	0.002	**0.018**
	IoU > 0.50|area = Medium	0.023	**0.175**	0.157	**0.220**	0.079	**0.196**	0.041	**0.208**

**Table 3 sensors-23-07185-t003:** Number of anomalies contained in each scene, as well as those identified using the two proposals evaluated for each of the sequences. The best results are marked in **bold**.

Model	Sequence	Methodology	Number of Anomalies	Number of Anomalies Detected
CenterNet HourGlass 104 Keypoints 1024 × 1024	Sequence 1	RAW	712	243
		OURS		**335**
	Sequence 2	RAW	370	4
		OURS		**84**
CenterNet HourGlass 104 1024 × 1024	Sequence 1	RAW	712	221
		OURS		**288**
	Sequence 2	RAW	370	60
		OURS		**123**
Faster R-CNN Inception ResNet V2 1024 × 1024	Sequence 1	RAW	712	57
		OURS		**222**
	Sequence 2	RAW	370	4
		OURS		**60**
EfficientDet D4	Sequence 1	RAW	712	12
		OURS		**217**
	Sequence 2	RAW	370	0
		OURS		**81**
SSD ResNet152 V1 FPN 1024 × 1024 (RetinaNet 152)	Sequence 1	RAW	712	4
		OURS		**208**
	Sequence 2	RAW	370	1
		OURS		**24**

## Data Availability

Not applicable.

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
