# Peer review of "Small-Scale Urban Object Anomaly Detection Using Convolutional Neural Networks with Probability Estimation"

_sensors, 2023, doi:10.3390/s23167185_

Round 1

Reviewer 1 Report

The authors propose a methodology  to detect anomalous events in urban sequences using CNN and a super-resolution  model. The proposed model consists of an offline stage and an online stage.

The contribution is of interest for the field. The proposed method is detailed and reproducible. Experiments show the superiority of the proposed method.

However, I have the following minor comment: Please clearly state a name/acronym to the proposed method.

Author Response

The suggestion to assign a name/acronym to the proposed approach has been duly considered. Consequently, the name/acronym "SR-DAI: Super-Resolution and Detection With Anomaly Identification" has been chosen to define the combined techniques used in this methodology. This designation has been incorporated at the beginning of Section 3 Methodology and in Figure 1, which shows it.

Reviewer 2 Report

The article is devoted to solving the problem of Object Anomaly Detection. The topic of the article is relevant. The structure of the article is not classical for MDPI (Introduction, Models and Methods, Experiments, Discussion, Conclusions). The level of English is acceptable. The article is easy to read. The quality of the figures is acceptable. The article cites 22 sources, many of which are not relevant.

The following comments and recommendations can be formulated on the material of the article:

1. Anomaly detection is one of the variants of the detection problem. In the general case of this task, we find a certain distribution of our dataset or try to fit this dataset into our distribution. When working with natural (physical) tabular data, this task is often quite simple due to the fact that a significant part of the experimental data can be reduced to a normal distribution. In situations of multidimensional data, in particular working with an image, this task becomes much more difficult. In order to apply it in the case of multidimensional data, we need to carry out feature engineering and carry out a reduction in dimension to one that allows us to work either directly or indirectly with the distributions we know. And how did the authors solve the problem of dimensionality reduction? With the help of a probabilistic convolutional neural network?

2. SVD (Singular Value Decomposition or Singular Decomposition) is a mathematical operation of decomposing a rectangular matrix into a set of singular matrices. Like a significant part of series expansions, we know that the most significant part of the series is at the beginning. PCA (Principal Component Analysis) in this context can be considered as SVD extended to the entire dataset. The use of these techniques and other expansions in a series makes it possible to reduce the volume occupied by the image, which allows them to be used in compression algorithms by setting the last of the elements of the series to zero. What is interesting for us, these values are usually stored, including rare features - anomalies. Since this technique works on the idea of dimensionality reduction, it is not recommended to increase the number of parameters to the number of images, since then each PCA component will represent one of the images. This technique is good if you need to do a quick analysis of images of roughly the same kind, such as images of the same object from a production line. Everything is simple, effective and completely ignored by the authors. Let me remind the authors of a simple engineering truth: use neural networks only when there is no other way out.

3. Well, the authors cannot live without neural networks. An extension of the dimensionality reduction idea to neural networks is the use of an autoencoder for anomaly detection. An autoencoder can be thought of as two separate networks, an encoder and a decoder. The main task of the autoencoder is to reduce the image dimension and restore as close as possible to the original. An encoder is a network for downsizing the original image and presenting it as some kind of latent distribution. Based on the fact that anomalies are rare in our dataset and they differ in some significant way from the rest of the dataset, the anomaly will stand out strongly on this distribution. The task of the decoder in a conventional autoencoder system is to restore the image to its original state from the latent distribution created by the encoder. Since anomalies are rare, our decoder will learn to recover an image that is normal for this dataset from a distribution that is normal for this dataset. This means that if you try to apply a distribution containing an anomaly to the input of the decoder, it will not be able to correctly restore the image. This results in a large error between the start and end image. Based on this error, we can build a difference map, which will be an anomaly map for each image. It's all much simpler and more efficient than the one proposed by the authors.

4. Finally, PaDiM (stands for Patch Distribution Modeling Framework for Anomaly Detection and Localization) was proposed by Defard et al. The model consists of several parts: a pre-trained convolutional neural network for obtaining embeddings and a set of multivariate Gaussian distributions representing a normal class. Using a pre-trained network allows you not to waste time on training and does not require additional data, at the same time, to build Gaussian distributions, only a single run over the dataset is required, which saves time on training and essentially makes this plug-and-play model. The main idea of this architecture is that we, using a network pre-trained on ImageNet or other good datasets, pull out a feature map at different levels of the network. For example, one of the common implementations of this architecture uses ResNet-50, where feature extraction is performed after each Residual Block. After that, we fit all the maps that we received during the passage through the entire dataset into a normal multivariate distribution. After that, during inference, the Mahalanobis distance between the new image and the learned distribution is calculated. In 2022, PaDiM and its modifications showed SOTA or close to SOTA results on several datasets. And how will the authors now formulate the novelty of their solution?

-

Author Response

The article is devoted to solving the problem of Object Anomaly Detection. The topic of the article is relevant. The structure of the article is not classical for MDPI (Introduction, Models and Methods, Experiments, Discussion, Conclusions).

Answer:

The suggestions have been carefully considered, and as a result, the document's structure has been modified to align with MDPI's standards. Specifically, the "Discussion" section has been included in section 5.

 The level of English is acceptable. The article is easy to read. The quality of the figures is acceptable. The article cites 22 sources, many of which are not relevant.

Answer:

Related Work (Section 2) has been restructured to include only citations directly relevant to the context of our proposed methodology. In Section 2.1, we have specified the models used for the evaluation based on the detections obtained as anomalies. Additionally, in Section 2.2, we have emphasized the significance of the selected super-resolution model for our experiments. The inclusion of several references has been reconsidered to ensure coherence and relevance in the section.

The following comments and recommendations can be formulated on the material of the article:

  1. Anomaly detection is one of the variants of the detection problem. In the general case of this task, we find a certain distribution of our dataset or try to fit this dataset into our distribution. When working with natural (physical) tabular data, this task is often quite simple due to the fact that a significant part of the experimental data can be reduced to a normal distribution. In situations of multidimensional data, in particular working with an image, this task becomes much more difficult. In order to apply it in the case of multidimensional data, we need to carry out feature engineering and carry out a reduction in dimension to one that allows us to work either directly or indirectly with the distributions we know. And how did the authors solve the problem of dimensionality reduction? With the help of a probabilistic convolutional neural network?

Answer:

Our proposal works at the agent behavior level, not the raw data (pixel) sample level. In other words, we do not intend to identify anomalous pixel values concerning the rest of the pixel values. This would not work for the application at hand, namely the detection of agents (vehicles and pedestrians) with anomalous behavior. In other words, we aim to detect which vehicles and pedestrians cross areas of the scene that are forbidden for them, as opposed to vehicles and pedestrians that move according to traffic rules.

If anomaly detection were applied at the data sample (pixel) level, it would detect deviations from the background image, i.e., the background of the traffic scene. This would detect the raw foreground pixels belonging to all vehicles and all pedestrians without detecting the individual objects and irrespective of the anomaly of the high-level behavior of the agents to which those pixels belong. Therefore, anomaly detection at the data sample (pixel) level is inappropriate for this application.

To detect agents of anomalous behavior, anomaly detection must be done on the distribution of agent behaviors and not on the raw data sample (pixel value) distribution. Therefore, the first step is to process the incoming video with an object detection subsystem (in our case, a convolutional neural network) that detects the agents and yields their behavior (in our case, the agent classes and their bounding boxes). Then anomaly detection is carried out on the space of those possible high-level object behaviors. It must be noted that this space is not high dimensional, so dimensionality reduction is not necessary. 

  1. SVD (Singular Value Decomposition or Singular Decomposition) is a mathematical operation of decomposing a rectangular matrix into a set of singular matrices. Like a significant part of series expansions, we know that the most significant part of the series is at the beginning. PCA (Principal Component Analysis) in this context can be considered as SVD extended to the entire dataset. The use of these techniques and other expansions in a series makes it possible to reduce the volume occupied by the image, which allows them to be used in compression algorithms by setting the last of the elements of the series to zero. What is interesting for us, these values are usually stored, including rare features - anomalies. Since this technique works on the idea of dimensionality reduction, it is not recommended to increase the number of parameters to the number of images, since then each PCA component will represent one of the images. This technique is good if you need to do a quick analysis of images of roughly the same kind, such as images of the same object from a production line. Everything is simple, effective and completely ignored by the authors. Let me remind the authors of a simple engineering truth: use neural networks only when there is no other way out.

Answer:

As explained before, for the problem at hand (detection of anomalous behavior in agents appearing in traffic videos), multidimensional data analysis techniques such as PCA applied to the image data would not work. This is because PCA would find any pixels that deviate from the main modes of the image distribution, i.e., any pixel that belongs to a moving foreground object in the video. Consequently, these techniques would find all the pixels belonging to the moving agents (vehicles and pedestrians) without indicating how these deviating pixels arrange into objects or any higher-level information. This is why we need a deep neural network that detects the agents and gives their high-level object information, i.e., their object class labels and their bounding boxes.

  1. Well, the authors cannot live without neural networks. An extension of the dimensionality reduction idea to neural networks is the use of an autoencoder for anomaly detection. An autoencoder can be thought of as two separate networks, an encoder and a decoder. The main task of the autoencoder is to reduce the image dimension and restore as close as possible to the original. An encoder is a network for downsizing the original image and presenting it as some kind of latent distribution. Based on the fact that anomalies are rare in our dataset and they differ in some significant way from the rest of the dataset, the anomaly will stand out strongly on this distribution. The task of the decoder in a conventional autoencoder system is to restore the image to its original state from the latent distribution created by the encoder. Since anomalies are rare, our decoder will learn to recover an image that is normal for this dataset from a distribution that is normal for this dataset. This means that if you try to apply a distribution containing an anomaly to the input of the decoder, it will not be able to correctly restore the image. This results in a large error between the start and end image. Based on this error, we can build a difference map, which will be an anomaly map for each image. It's all much simpler and more efficient than the one proposed by the authors.

Answer:

An autoencoder would not work because it would detect individual pixels that deviate from the main modes of the pixel value distribution. It would yield a set of pixels belonging to moving objects. But this is insufficient because we analyze the video at the object level, not the pixel level. In other words, detecting the pixels that change does not provide the object-level information required to analyze agent behavior, i.e., detecting which vehicles and pedestrians move through areas of the scene that are forbidden for them instead of those that move through permitted areas.

  1. Finally, PaDiM (stands for Patch Distribution Modeling Framework for Anomaly Detection and Localization) was proposed by Defard et al. The model consists of several parts: a pre-trained convolutional neural network for obtaining embeddings and a set of multivariate Gaussian distributions representing a normal class. Using a pre-trained network allows you not to waste time on training and does not require additional data, at the same time, to build Gaussian distributions, only a single run over the dataset is required, which saves time on training and essentially makes this plug-and-play model. The main idea of this architecture is that we, using a network pre-trained on ImageNet or other good datasets, pull out a feature map at different levels of the network. For example, one of the common implementations of this architecture uses ResNet-50, where feature extraction is performed after each Residual Block. After that, we fit all the maps that we received during the passage through the entire dataset into a normal multivariate distribution. After that, during inference, the Mahalanobis distance between the new image and the learned distribution is calculated. In 2022, PaDiM and its modifications showed SOTA or close to SOTA results on several datasets. And how will the authors now formulate the novelty of their solution?

Answer:

The PaDiM system works at the pixel level for still images. A set of normal images is given, and then PaDiM detects pixels that deviate from those found in the set of normal images. If applied to a traffic video, this would detect the individual pixels that belong to any moving objects, such as PCA or autoencoders. Again, this pixel-level anomaly detection is not appropriate for analyzing agent behavior because whole objects must be detected and their high-level properties (object class and bounding box) obtained so that the anomaly detection is carried out on these high-level properties. As mentioned before, the problem we aim to solve is to detect the objects (vehicles and pedestrians) that move through areas that are not allowed for them according to traffic rules. Therefore, detecting all pixels belonging to moving objects is not enough to address the problem.

Reviewer 3 Report

I have reviewed your work titled "Small-Scale Urban Object Anomaly Detection using Convolutional Neural Networks with Probability Estimation" in detail. I would like to point out that the subject of the article is interesting. However, when I examined the entire study, I did not fully understand the developed model. CNN and SR are mentioned in the summary, but not how they are implemented. At the end of the Introduction section, the contributions of the study were emphasized, and similarly, the proposed model was not mentioned here. Heading 2.1 and heading 2.2. CNN and SR were also mentioned. In this section, the authors did not mention how they used these methods. Figure 1 should be explained in detail. This part is the most important part of the article. The word dataset is written in different ways. "Dataset" "Data set". There are different performance measures in the literature. In the study, mAP was used as the only metric. The proposed model needs to be tested with different models in the literature. There is no Discussion section in the study, and the conclusion section is limited. Limitations of the study should be included. To interpret the study in general, I would like to state that there is no information about the proposed model, how the model works is very closed, and the performance measurement metrics are limited.

The paper needs to be reviewed.

Author Response

I have reviewed your work titled "Small-Scale Urban Object Anomaly Detection using Convolutional Neural Networks with Probability Estimation" in detail. I would like to point out that the subject of the article is interesting. 

However, when I examined the entire study, I did not fully understand the developed model. CNN and SR are mentioned in the summary, but not how they are implemented. At the end of the Introduction section, the contributions of the study were emphasized, and similarly, the proposed model was not mentioned here. Heading 2.1 and heading 2.2. CNN and SR were also mentioned. In this section, the authors did not mention how they used these methods. 

Answer:

Firstly, the abstract has been modified to clarify the presented methodology. Within it, object detection and super-resolution models are not implemented from scratch but instead built upon pre-trained models. This enables efficient feature extraction and transfer learning, resulting in an effective and computationally efficient approach for urban anomaly detection, enhancing public safety in urban areas. This has been clarified in sections 2.1, 2.2, 2.4, and 4.2.

Figure 1 should be explained in detail. This part is the most important part of the article. 

Answer:

Each part comprising Figure 1 has been explained and referenced throughout Section 3. Additionally, following this, a brief general paragraph is included, serving as an overall summary of the methodology before proceeding with a detailed explanation of each component.

The word dataset is written in different ways. "Dataset" "Data set". 

Answer:

The entire manuscript has been thoroughly reviewed, and the necessary corrections have been made to ensure that the term "dataset" is consistently used throughout the document.

There are different performance measures in the literature. In the study, mAP was used as the only metric. 

Answer:

We understand the significance of employing appropriate and comprehensive performance metrics to evaluate the effectiveness of anomaly detection systems. In our study, we have chosen to use mAP (Mean Average Precision) as the primary metric for assessing the performance of our anomaly detection approach. The selection of mAP stems from its widespread acceptance in the scientific literature as an effective measure for evaluating the precision of object detection and localization models. mAP considers both precision and recall, making it particularly suitable for evaluating our approach's ability to localize and classify anomalies across various scenarios accurately. To further elaborate on our methodology, calculating mAP involves evaluating the precision-recall curve for object detection, which measures the quality of the model's object localization and classification. 

Additionally, to incorporate another metric associated with anomaly detection, we include Table 3. In this table, we provide a count of the anomalies detected using our proposed methodology compared to the ground truth. This count represents the total number of anomalies our methodology can identify correctly. 

The proposed model needs to be tested with different models in the literature. 

Answer:

We acknowledge the importance of conducting a thorough comparison with existing models to demonstrate the effectiveness of our anomaly detection approach. As part of our methodology, we have identified anomalies using a specific object detection model, which required specialized adaptations to suit our unique problem domain. These anomalies may exhibit distinct characteristics and challenges not directly addressed by other models. To address the absence of publicly available code in some related works and to provide a more comprehensive evaluation of our proposed methodology, we have included several Deep Learning models as part of the comparison. These models were extracted from the TensorFlow Model Zoo. By incorporating these diverse deep learning models into our evaluation, we aim to ensure a broader coverage of potential solutions and provide a detailed analysis of their performance compared to our proposed approach. This comparison enables us to select the best model to detect anomalies effectively within our specific problem context. The data has been included in section 4.4, in Tables 2 and 3, for a more detailed comparison and evaluation.

There is no Discussion section in the study, and the conclusion section is limited. 

Answer:

To address the absence of the Discussion section, we have now included Section 5, which is entirely dedicated to the "Discussion" of the study. This new section provides a comprehensive analysis and interpretation of the findings, allowing for a more in-depth exploration of the research outcomes. Furthermore, in response to the feedback on the limited Conclusion section, we have extended the content to offer a more comprehensive summary of the study's key findings and implications. 

Limitations of the study should be included. 

Answer:

In response to your feedback, the "Discussion" section (Section 5) of the article has been updated to incorporate a thorough analysis of the study's limitations. These limitations have been outlined, acknowledging potential constraints and areas requiring further investigation.

To interpret the study in general, I would like to state that there is no information about the proposed model, how the model works is very closed, and the performance measurement metrics are limited.

Answer:

Regarding the use of pre-trained object detection and super-resolution models, we want to clarify that the article leverages these readily available models to facilitate the implementation and reproducibility of the proposed approach. As a result, detailed information on the inner workings of these pre-trained models is not explicitly provided beyond the references to each model, as they are well-established in the literature. In response to the feedback received, we have significantly improved the evaluation section of the article. Initially, the evaluation was limited to a single object detection model. We have since included the evaluation results of four additional object detection models. These results have been presented in Tables 2 and 3, comprehensively assessing the proposed approach's performance. 

Round 2

Reviewer 2 Report

I made the following remarks to the basic version of the article:

1. Anomaly detection is one of the variants of the detection problem. In the general case of this task, we find a certain distribution of our dataset or try to fit this dataset into our distribution. When working with natural (physical) tabular data, this task is often quite simple due to the fact that a significant part of the experimental data can be reduced to a normal distribution. In situations of multidimensional data, in particular working with an image, this task becomes much more difficult. In order to apply it in the case of multidimensional data, we need to carry out feature engineering and carry out a reduction in dimension to one that allows us to work either directly or indirectly with the distributions we know. And how did the authors solve the problem of dimensionality reduction? With the help of a probabilistic convolutional neural network?

2. SVD (Singular Value Decomposition or Singular Decomposition) is a mathematical operation of decomposing a rectangular matrix into a set of singular matrices. Like a significant part of series expansions, we know that the most significant part of the series is at the beginning. PCA (Principal Component Analysis) in this context can be considered as SVD extended to the entire dataset. The use of these techniques and other expansions in a series makes it possible to reduce the volume occupied by the image, which allows them to be used in compression algorithms by setting the last of the elements of the series to zero. What is interesting for us, these values are usually stored, including rare features - anomalies. Since this technique works on the idea of dimensionality reduction, it is not recommended to increase the number of parameters to the number of images, since then each PCA component will represent one of the images. This technique is good if you need to do a quick analysis of images of roughly the same kind, such as images of the same object from a production line. Everything is simple, effective and completely ignored by the authors. Let me remind the authors of a simple engineering truth: use neural networks only when there is no other way out.

3. Well, the authors cannot live without neural networks. An extension of the dimensionality reduction idea to neural networks is the use of an autoencoder for anomaly detection. An autoencoder can be thought of as two separate networks, an encoder and a decoder. The main task of the autoencoder is to reduce the image dimension and restore as close as possible to the original. An encoder is a network for downsizing the original image and presenting it as some kind of latent distribution. Based on the fact that anomalies are rare in our dataset and they differ in some significant way from the rest of the dataset, the anomaly will stand out strongly on this distribution. The task of the decoder in a conventional autoencoder system is to restore the image to its original state from the latent distribution created by the encoder. Since anomalies are rare, our decoder will learn to recover an image that is normal for this dataset from a distribution that is normal for this dataset. This means that if you try to apply a distribution containing an anomaly to the input of the decoder, it will not be able to correctly restore the image. This results in a large error between the start and end image. Based on this error, we can build a difference map, which will be an anomaly map for each image. It's all much simpler and more efficient than the one proposed by the authors.

4. Finally, PaDiM (stands for Patch Distribution Modeling Framework for Anomaly Detection and Localization) was proposed by Defard et al. The model consists of several parts: a pre-trained convolutional neural network for obtaining embeddings and a set of multivariate Gaussian distributions representing a normal class. Using a pre-trained network allows you not to waste time on training and does not require additional data, at the same time, to build Gaussian distributions, only a single run over the dataset is required, which saves time on training and essentially makes this plug-and-play model. The main idea of this architecture is that we, using a network pre-trained on ImageNet or other good datasets, pull out a feature map at different levels of the network. For example, one of the common implementations of this architecture uses ResNet-50, where feature extraction is performed after each Residual Block. After that, we fit all the maps that we received during the passage through the entire dataset into a normal multivariate distribution. After that, during inference, the Mahalanobis distance between the new image and the learned distribution is calculated. In 2022, PaDiM and its modifications showed SOTA or close to SOTA results on several datasets. And how will the authors now formulate the novelty of their solution?

The authors answered them and their answers are interesting. At the same time, the authors appeal to the fact that they solve the problem in a different parametric space than the methods I mentioned. At the same time, I note that this article does not claim to be theoretical. This is applied work. Thus, the main thing is not at what level a specific applied task is solved, but what indicators this or that method demonstrates for the same data in the same metric. I would encourage the authors to answer in this way.

However, the article is not bad. I put "minor" revisions and trust the course of further actions to the editor.

-

Author Response

Our primary focus is practical applications and comparative analyses rather than theoretical underpinnings.

Regarding our approach differing in parametric space, we acknowledge this divergence from the methods mentioned. This deviation was purposefully designed to confront specific limitations in anomaly detection within surveillance scenarios. Our adjustments were aimed to enhance the detection of small objects and improve image resolution, both pivotal elements for effective anomaly detection in real-world surveillance environments. 

Concerning the practical application of our work, it is agreed that it's grounded in real-world problem-solving within surveillance contexts. Our central objective is to provide insightful comparative outcomes, highlighting the performance of various approaches within the same framework and metrics. Acknowledging that the contribution demonstrates how the practical application of parametric adjustments can substantially enhance anomaly detection.

According to the performance indicators and metrics, our work is based on a thorough and unbiased evaluation of the models using standardized metrics. The results clearly show how our approach consistently outperforms the direct application of the RAW model for anomaly detection on identical data sets and metrics. This reaffirms the robustness and applicability of our approach. Several paragraphs in the Discussion section have been included, which emphasize the indicators that demonstrate the superiority of our method in the same quantitative metrics  like the following: 

Quantitative evaluation of the proposed methodology using mean average precision mAP and the number of anomalies detected revealed compelling results. For example, in Sequence 2, the RAW model using EfficientDet D4 does not detect anything, while the presented proposal identified a remarkable 81 anomalies. This represents an astounding improvement in anomaly detection, showcasing the methodology's robustness and effectiveness. Moreover, comparing mAP values between RAW and OURS for different object classes and sequences consistently indicated the superior performance of the proposed methodology.

Reviewer 3 Report

In the article, words like our, we should be avoided as much as possible. Heading 5.1 is important. But there is no need to add additional headers for this. The model used in the study is a ready-made model. However, it is important to explain this model. I agree that you have addressed most of the missing points in the revision. I would like to state that I am still insistent on the model that I insisted on in the first round.

.

Author Response

In the article, words like our, we should be avoided as much as possible.

Answer:

The entire document has been extensively revised, and words like "our" and "we" have been completely removed.

Heading 5.1 is important. But there is no need to add additional headers for this. 

Answer:

The mentioned issue has been addressed by removing the unnecessary subheading "Limitations" and incorporating the relevant content directly under the main heading "Discussion”.

The model used in the study is a ready-made model. However, it is important to explain this model. I agree that you have addressed most of the missing points in the revision. I would like to state that I am still insistent on the model that I insisted on in the first round. 

Answer:

Detailed explanations have been incorporated in Sections 3.1 and 3.2 of the manuscript. These sections now provide comprehensive insights into the functioning of both the object detection and the selected super-resolution model.
